Design and implementation of the international news commentary Data Intelligent Processing System

Liao Shiting 1
Wang Yunpei 2
Wang Qingnian qnwang@scut.edu.cn 1 3
1 School of Journalism and Communication, South China University of Technology , Guangzhou , Guangdong , China
2 School of Economics and Management, South China Normal University , Guangzhou , Guangdong , China
3 School of International Education, South China University of Technology , Guangzhou , Guangdong , China
Pires Ivan Miguel
Electronic publication date: 2024 Oct 9
Publication date: 2024
Volume: 10
Electronic Location ID: e2376
Received 2023 Dec 1; Accepted 2024 Sep 10
Copyright: ©2024 Liao et al.
Copyright year: 2024
Copyright holder: Liao et al.
License: This is an open access article distributed under the terms of the Creative Commons Attribution License, which permits unrestricted use, distribution, reproduction and adaptation in any medium and for any purpose provided that it is properly attributed. For attribution, the original author(s), title, publication source (PeerJ Computer Science) and either DOI or URL of the article must be cited.
License URL: https://creativecommons.org/licenses/by/4.0/

Keywords: International news commentary, Intelligent processing, Data analysis, Visualization

Funding: The National Social Science Fund 21BSH097 The Key Project of the Center of Sino-Foreign Language Cooperation & Exchange (2021), Ministry of Education, China This study was funded by the National Social Science Fund (No: 21BSH097), China, the Key Project of the Center of Sino-Foreign Language Cooperation & Exchange (2021), Ministry of Education, China. The funders had no role in study design, data collection and analysis, decision to publish, or preparation of the manuscript.

==============================
In the digital media age, international news commentary has changed, creating challenges such as information overload and noise. Traditional platforms often need more data-driven analysis capabilities. This study presented a specialized intelligent system for processing international news commentary data. The system provided robust analysis tools, automated recommendations, and summarization capabilities. Its comprehensive modules included data crawling, processing, visualization, and retrieval. Experimental results demonstrated the system’s effectiveness in processing data and generating valuable insights. Users were able to gain objective insights into topics, emotions, and dissemination patterns. The system provided valuable resources for communication research, enhancing theoretical understanding and practical applications in the field.

Introduction

The emergence of digital media has had a profound impact on international news commentary, which has experienced unprecedented explosive growth. This rapid digital development underscores the shortcomings of traditional news platforms in keeping pace with these advancements. Traditional news platforms like news websites, online news portals, and aggregators typically provide basic search and categorization functionalities. However, they often need more in-depth data-driven analytical capabilities, such as automated content summarization or complex text analysis. These platforms’ limitations lie in their primary focus on surface-level content processing rather than delving into deeper information or insights.

Although various online analytical systems are currently available and applicable in social media monitoring, market research, and sentiment analysis, they generally offer functionalities like sentiment analysis, topic identification, natural language processing, and user behavior analysis. However, these systems tend to employ a more generic methodology rather than being specifically designed to address the nuances and specificities of international news commentary. Consequently, while these existing tools are helpful in certain respects, they often need to fully meet the specialized requirements of international news commentary analysis.

As a genre of journalism and political discourse, international commentary involves expressing opinions and elucidating viewpoints and stances directly related to important or interesting global issues (Zhao, 2014). In particular, international news commentary has been crucial in providing comprehensive explanations and analyses of international events and issues (Dong & Shao, 2017). By presenting both information and viewpoints, China can effectively introduce its policies, culture, and values, enhancing its ability to communicate its voice and maintain a positive international image. However, international communication still faces the dilemma of “having reasons but not being able to express them”, “expressing them but not disseminating them widely”, and “disseminating them but not making an impact” (Shi, 2021). Furthermore, how to continuously and effectively demonstrate China’s goodwill to the world, enabling international audiences to understand the real, multifaceted, and comprehensive nature of China, and eliminate the misunderstandings and prejudices of some countries, is an urgent issue for international communication to address (Shen & Jin, 2023).

Due to the extensive, diverse, and complex nature of international news commentary data, users face information overload, cultural barriers, and comprehension difficulties. Effectively extracting valuable information from this data and enabling users to quickly understand and digest it is a significant challenge in news analysis. In addition, traditional methods of collecting and processing information one at a time waste significant human resources and are cumbersome and time-consuming. Moreover, the abundance of complex textual data hinders users’ understanding and assimilation, making it inconvenient to select, analyze, and integrate international news commentary.

The system designed in this study effectively alleviated the above problems and challenges. The system organized, computed, and analyzed extensive and multi-layered text data through intelligent processing techniques, helping users to extract valuable information from commentary data, including emotions, perspectives, and trends. The system also presented complex data intuitively using charts, graphs, and other visual formats, enhancing user understanding and lowering the barrier to entry, thereby expanding its reach. In addition, the system made international news commentary more readable and understandable than traditional news websites by integrating intelligent processing techniques. It provided users with a better reading experience and significantly improved global communication efficiency.

This research aimed to accurately crawl, process, and visualize international news commentary data to improve global communication efficiency. This study developed a web crawler program using Python and integrated natural language processing algorithms and third-party libraries to implement functions such as summarization, topic analysis, keyword extraction, sentiment analysis, similarity recommendation, and information retrieval. In addition, this study used the Flask framework and the Echarts chart library to create an intuitive and user-friendly interface that allowed users to understand the results of news data processing visually. The main contributions of this study included, but were not limited to:

•Precise data processing. This study designed and implemented an efficient web crawler program to ensure the comprehensive collection and accuracy of international news commentary data.

•Multi-layered text analysis. Natural language processing algorithms were used to conduct in-depth analysis of the commentary data, including summarization, topic analysis, keyword extraction, and sentiment analysis. These analyses deepened the understanding of the content and viewpoints expressed in the commentary.

•Intuitive visual representation. The Flask framework and the Echarts library were used to create an intuitive and user-friendly interface. Complex data was presented in charts and graphs, allowing users to understand and analyze the results of news data processing quickly.

Through these advanced technologies and interface designs, this study provided a viable solution for processing and disseminating international news commentary data. It promoted cross-cultural communication and understanding, enhanced user experience and engagement, and provided valuable decision support for decision-makers. The research had significant practical value and societal significance.

Related Works

The section was divided into subsections to provide a focused overview of each key area pertinent to this research. This division facilitated a clearer understanding of the existing literature in web crawling, sentiment analysis, and data visualization and how they related to our study. The segmentation was designed to systematically present the background, enabling a comprehensive analysis of the state of research at the time and identifying gaps our work aimed to address.

Web crawling

Since the 1990s, web crawling technology has continuously evolved (Bergmark, Lagoze & Sbityakov, 2002), matured, and found widespread applications in various fields such as data mining, public opinion analysis, and market research. With the advancement of big data and artificial intelligence, web crawling technology has shown trends toward greater intelligence, adaptability, and specialization. In addition, there is a growing trend toward broader and deeper applications of web crawlers.

A web crawler is an automated program or script designed to retrieve information from web pages. Its basic principles are similar to how ordinary users browse web pages using web browsers (Cho, 2001). First, based on the characteristics of the target to be crawled, valuable URLs are selected as seed URLs, which define the crawling scope of the web crawler. Starting from these seed URLs, the crawler sends HTTP requests to the servers to retrieve the web page source codes. It then parses the web page information, extracts the required data, and stores it for future use. At the same time, the crawler adds the crawled pages to the list of previously visited URLs. While analyzing web pages, the crawler extracts other URLs from the web pages. These URLs are compared with the list of previously visited URLs. If a URL is not in the list, it is added to the queue of URLs to be crawled. The crawler continues to extract URLs from the queue, repeating the above steps until the queue is empty or a termination condition is met. Specifically, the basic workflow of a generic Web crawler is as follows (as shown in Fig. 1):

Figure 1 General web crawling workflow.

•Determine the Target Website and Starting Page. A Web crawler must specify the Web site it wants to crawl and the start page. Once the target site is identified, the crawler sends an HTTP request to that site to retrieve its web page content.

•Send HTTP requests to obtain page content. To retrieve data from a Web site, the Web crawler must simulate the behavior of a browser. It does this by sending HTTP requests to the server and receiving the web page content that the server sends back. Python tools such as the Requests library and the Scrapy framework are commonly used to send HTTP requests.

•Parse the page content. The web page content that the crawler receives is usually in HTML format. The necessary information must be extracted through parsing. Parsing tools such as Python’s BeautifulSoup library, XPath, and regular expressions are commonly used. After parsing, the processed page is added to the list of previously visited URLs to facilitate deduplication in the subsequent process.

•Recursively crawl URLs in the queue. A web page can contain multiple links. Based on user-defined criteria, the crawler extracts the desired links. These selected URLs are compared with those in the list of previously visited URLs to remove duplicates. The deduplicated URLs are then added to the queue of URLs to be crawled. The crawler continues this process recursively until the queue of URLs to be crawled is empty or a termination condition is met.

•Data storage. Crawled data can be stored in various formats, including local files and databases. Standard data storage methods include CSV files, JSON files, and MySQL databases.

Often, multiple links within a web page may be nested and duplicated (Yan & Garcia-Molina, 1995). Without comparing the URLs fetched by the crawler with those already in the visited queue, there is a risk of redundant crawling of the same page. This redundancy not only reduces the efficiency of the web crawler but can also lead to infinite loops. Therefore, implementing an effective deduplication mechanism is crucial for developing a robust web crawler (Agarwal et al., 2009). Web page deduplication refers to the process of avoiding redundant crawling of the same web page during the crawling process. This process is essential to conserve crawling resources and increase efficiency (Wu, Chen & Ma, 2003). In this research, web page deduplication focused primarily on URL deduplication. The underlying principle was to mark the URLs crawled during the spidering process and filter out unnecessary and duplicate URLs.

Sentiment analysis

Sentiment analysis uses natural language processing and text mining to identify and extract subjective information from text (Hussein, 2018). Three mainly used approaches for sentiment analysis include sentiment analysis based on emotional dictionary, sentiment analysis based on traditional machine learning, and sentiment analysis based on deep learning (Wang & Yang, 2021).

The key to the method based on sentiment lexicons lies in constructing the sentiment lexicon. Alfreihat et al. (2024) developed an Emoji Sentiment Lexicon (Emo-SL) for Arabic tweets and enhanced sentiment classification by integrating emoji-based features with machine learning techniques. Hutto & Gilbert (2014) introduced ‘A Parsimonious Rule-based Model for Sentiment Analysis of Social Media Text’, known as VADER. They employed a rule-based and minimalist model to assess the sentiment of tweets and found that VADER outperformed individual human raters. Given the nature of international news commentary, this research used sentiment analysis based on the VADER emotional dictionary as a suitable and convenient approach.

Visualization

With the continuous expansion of data volumes, the results of data analysis are becoming increasingly large and complex. In this context, visualization is a practical approach for elucidating and presenting large data sets (Meng & Ci, 2013). Card, Mackinlay & Shneiderman (1999) asserted that information visualization involves the structural transformation of data into visually perceptible attributes within graphical representations. This process utilizes computational, interactive, and visual modes of data representation to enhance cognitive abilities and optimize the effectiveness of data analysis, thereby helping users achieve a deeper understanding and analysis of complex datasets.

Echarts

The implementation of data visualization can be broadly categorized into two primary methodologies: one is user-oriented, such as Dycharts and Tableau. These tools provide intuitive user interfaces that make creating visual charts and graphs easy and quickly. The other type is developer-oriented and is represented by third-party libraries such as Matplotlib in Python and JavaScript-based visualization frameworks such as D3.js, Highcharts, ECharts, and Chart.js (Pei, 2018). With increased adaptability and customization, these frameworks allow developers to add dynamic effects and interactive features to web pages.

ECharts, developed and maintained by Baidu, is a JavaScript library for constructing interactive charts and visualizations. It includes several popular chart types, ranging from line and bar charts to scatter plots and pie charts. In addition, ECharts provides robust support for interactive operations and real-time data updates, contributing to a dynamic and immersive user experience (Li et al., 2018).

Flask

Flask is a lightweight web application framework based on Python (Copperwaite & Leifer, 2015). Its notable features include flexibility, user-friendly interface, and scalability, supported by the foundational components of Werkzeug and Jinja2 (Relan , 2019). As a result, Flask is optimal for rapid prototyping, data visualization, and compact web environments. This inherent ease of use allows developers with limited experience in web development to quickly orchestrate websites with increased proficiency (Niu & Li, 2019). Classified as a “microframework”, Flask is characterized by its concise core codebase and streamlined architectural essence. Unlike its counterparts, such as Django and Pyramid, among Python web frameworks, Flask is distinguished by its lightweight nature and adaptive versatility (Ghimire, 2020). The underlying design ethos of the Flask framework is rooted in a desire to maintain an unadorned elegance while preserving a nuanced degree of adaptability.

Intelligent processing system

Intelligent processing systems refer to tools that use intelligent processing techniques for in-depth analysis and efficient operations on data to provide users with intuitive, efficient, and accurate user experiences. Significant progress has been made in intelligent news processing systems.

Current research on intelligent news processing systems is focused on several key areas. One such area is news recommendation systems, which leverage user behavioral data to customize content. Liu et al. (2024) developed and tested the News Recommendation with Attention Mechanism (NRAM) model, an advanced system that uses attention-based mechanisms to personalize news recommendations. Their research demonstrated that the NRAM model significantly outperforms traditional models by effectively integrating multi-head self-attention and additive attention, thereby enhancing the accuracy and personalization of news content for users on digital platforms.

Another critical area is fake news detection systems, dedicated to identifying and mitigating the spread of false and misleading information to enhance the credibility of news sources. Scholars have employed intelligent techniques to improve information reliability and credibility. Kapusta, Drlik & Munk (2021) showed that applying morphological analysis using part-of-speech (POS) tags and n-grams significantly improved the accuracy of fake news detection. Similarly, Agarwal et al. (2022) introduced a novel method that utilizes spatiotemporal features for fake news classification, achieving promising results in domain-specific COVID-19 fake news prediction and highlighting the importance of spatial and temporal features in fake news detection.

Additionally, news feature mining systems are engineered to extract essential information and identify trends from vast datasets. Xie & Wang (2023) developed an innovative news feature mining system by integrating the Internet of Things with artificial intelligence (AI). This system effectively and accurately processes news data in the rapidly expanding digital environment.

Lastly, sentiment and opinion mining systems are employed to evaluate the sentiments and public opinions embedded within news content. Some scholars have used data mining and data analysis techniques to extract opinions and emotions from financial news, building models for market forecasting, predicting future stock prices (Mehta, Pandya & Kotecha, 2021), and predicting bitcoin price movements (Gurrib & Kamalov, 2022).

In summary, although the research field of news intelligent processing systems has achieved rich results, the development of systems for the specific genre of international news commentary is still relatively scarce. Unlike general news reporting, international news commentary primarily focuses on articulating opinions and stances. In view of this, the paper proposes a specialized intelligent processing system for international news commentary, developed through an extensive review of existing research. The following section discusses in detail the design and implementation of the system, examining how it effectively addresses the specific needs of international news commentary.

Methods

This article introduced an intelligent processing system for international news commentary data that addressed the complexity and ambiguity inherent in the emotional context of international news commentary. The system utilized web crawling techniques to collect, process, and store relevant international news commentary information in a database. Additionally, it allowed users to perform intricate data retrieval operations, fulfilling the need for fast and accurate information retrieval and mitigating issues related to information overload and dispersion. The system employed intelligent processing techniques to conduct statistical analyses and in-depth evaluations of the diverse and vast international news commentary data. It assisted users in extracting valuable insights from multi-layered textual data, including sentiments, opinions, and trends.

Furthermore, the system employed intelligent methodologies to analyze international news commentary data, providing a viable solution for processing and disseminating data and enhancing user experience and engagement. It promoted cross-cultural communication and understanding. Moreover, this system comprehensively organized and deeply analyzed data from the last six years, offering researchers a macroscopic perspective and a broader research scope. This expanded the breadth and depth of international news commentary research. The system leveraged visualization techniques, presenting complex data through images and charts, aiding users in better understanding and analyzing the data and offering robust support for decision-making. The system technical architecture diagram is illustrated in Fig. 2, which delineates the multi-tiered structure of the system.

Figure 2 System technical architecture diagram.

The Intelligent Processing System for International News Commentary Data comprised four sub-modules: International News Commentary Data Crawling Module, International News Commentary Data Processing Module, International News Commentary Data Visualization Display Module, and International News Commentary Data Retrieval Module.

The crawling module was implemented in Python, using web scraping techniques with the XML parser. It used Redis hash tables to deduplicate URLs, employed proxies to bypass anti-crawling mechanisms, adhered to web page crawling intervals based on update patterns, and stored data in a database. This module provided real-time retrieval and presentation of international news commentary data on the website.

Implemented in Python, this data processing module utilized third-party libraries and built-in modules for statistical analysis, computation, and further processing of collected textual data. It included abstract analysis based on the TextRank algorithm, sentiment analysis using the VADER sentiment lexicon, and keyword extraction using nltk. probability, topic analysis utilizing the gensim corpora module and LDA algorithm, and similarity-based recommendations using cosine similarity algorithms.

The data visualization display module, implemented using the frontend Echarts plugin and the backend Flask framework, retrieved the processed results from the database and presented them on the web page using various charts and graphs, including word clouds, pie charts, and bar graphs.

Built on top of the Flask web framework and using Ajax technology for front-to-back interaction, the data retrieve module supported keyword searches and provided multiple search criteria to help users quickly and accurately find the desired content within the vast information. In Fig. 3, the international news commentary data intelligent processing system mainly included modules for data crawling, data processing, visualization display, and data retrieval.

Experiment and Results

The intelligent processing system for international news commentary data was implemented using Python, allowing for automatic and dynamic incremental data collection. Python’s built-in modules and third-party libraries were used for data processing and analysis. Additionally, the system was integrated with the Flask framework and the Echarts plugin to present the processing results as charts on a webpage (as shown in Fig. 4). This section explained the implementation process for the following functionalities based on the design: data crawling, data processing, data visualization, and data retrieval.

Figure 3 System functional architecture.

Figure 4 Home page.

Implementation of data crawling function

The Webpage Download and Parsing Module was responsible for establishing connections with both Redis and MySQL databases. Using proxies and random user agent information, this module captured web page data from specified URLs using the requests library. During the parsing of webpage content, XPath (Clark & De Rose, 1999) expressions were employed. The results of XPath operators were Document Object Model (DOM) objects. These expressions aided in pinpointing target nodes within the webpage, allowing for the precise extraction of crucial information such as titles, timestamps, index image URLs, detail page URLs, and main text content. The flexibility and precision of XPath ensured that the system could accurately locate and extract the required field data.

The Webpage Information Update Module checked the Redis database for the presence of each article’s detail page URL during processing. If the URL was not found in the database, indicating a new article, the system stored its information in both the MySQL and Redis databases. However, if the detail page URL existed in the database, the system flagged the article as duplicate information, preventing the redundant storage of identical data. The seamless integration of these two modules ensured that the system could automatically and efficiently extract news information from web pages. It also enabled real-time updates to the database, ensuring the timeliness and accuracy of the data.

The data pre-processing steps implemented in the Data Pre-processing Module include:

•Removing extra spaces and newlines. Within the get_first_text () function, the strip() method stripped leading and trailing spaces and newlines from strings. This ensured that the extracted text content was clean, avoiding unnecessary spaces and newlines that could affect text processing and analysis.

•Handling of date and time formats. Time information extracted from web pages may contain extra spaces or other characters. Cleaning operations were applied to ensure the consistency and standardization of time formats. This benefited subsequent time series analysis, statistical analysis, and other operations.

•Dealing with special characters. Webpage text data may contain special characters, such as non-English letters and symbols. These special characters were either escaped or removed during data preprocessing to ensure data accuracy and usability. The presence of special characters can interfere with text analysis and machine learning model training, so cleaning these characters’ making was crucial.

•Handling missing data. Information retrieved from web pages may have specific fields that were empty. During data preprocessing, these missing values were detected, and measures were taken to fill in the gaps, allowing for further analysis and modeling. The system used “N/A” or other specific strings to identify missing data, which helped with detection and handling during subsequent processing.

These data preprocessing steps ensured data consistency, accuracy, and usability, providing a reliable data foundation for subsequent data analysis and modeling processes.

Implementation of data processing function

The section details the specific methods and technologies used for processing data within the system. It includes utilizing advanced techniques like natural language processing and data visualization, essential for efficiently managing and interpreting large volumes of international news commentary data. The section outlines the systematic procedures of data acquisition, analysis, processing, and presentation, emphasizing the transformation of news data into actionable insights and structured information, which is crucial for understanding the system’s operational framework.

Implementation of abstract analysis

This section implemented text summarization for international news commentary using the TextRank algorithm (as shown in Fig. 5). The specific steps were as follows:

The vectorizer. fit(contents) step involved using a TF-IDF vectorizer to transform the text into feature vectors, which is suitable for sentence similarity calculations. TF-IDF (Term Frequency-Inverse Document Frequency) is a statistical method used to evaluate the importance of a word within a corpus. The TF-IDF algorithm consists of term frequency (TF) and inverse document frequency (IDF). TF assesses the prevalence of a term within a document, reflecting its importance. The calculation of TF is as follows: (1) TFt,d=NumberoftimestermtappearsindocumentdTotalnumberoftermsindocumentd.

Equation (1) represents the frequency of occurrence of a word within a single document, with the TF value increasing as the frequency of occurrence becomes higher.

Inverse Document Frequency (IDF) measures a term’s rarity across the entire corpus, offering a counterbalance to the TF. The computation of IDF is detailed as follows: (2) IDFt,D= logTotalnumberofdocumentsinthecorpusDNumberofdocumentscontainingtermt+1.

Equation (2) demonstrates that the IDF (Inverse Document Frequency) value is inversely related to the prevalence of a word throughout the corpus. A lower occurrence frequency signifies a higher degree of uniqueness for the word, resulting in an increased IDF value.

Figure 5 Abstract analysis.

The final TF-IDF score is obtained by multiplying TF and IDF, as presented: (3) TF−IDFt,d,D=TFt,d∗IDFt,D.

Equation (3) represents that a higher TF-IDF score indicates a term’s significant presence in a particular document while being rare across the corpus, thus highlighting its importance for the document’s content.

The preprocess_text(text) function was crucial for text preprocessing, where it removed HTML tags using regular expressions and processed the text through tokenization and cleaning. This step ensured that the sentences were standardized for further processing.

Subsequently, the text_rank_summary(text, num_sentences =2) function generated summaries by preprocessing text into clean sentences, vectorizing them, and constructing a sentence similarity matrix. The application of the PageRank algorithm was vital in determining sentence scores and selecting the top-ranked sentences for the summary.

Finally, the page_rank(similarity_matrix, max_iterations =100, tolerance =1e−4, damping_factor = 0.85) function implemented the PageRank algorithm to iteratively compute sentence scores, continuing until reaching the maximum number or the tolerance level, resulting in the final sentence scores. These steps collectively formed a comprehensive process for analyzing and summarizing text data.

The TextRank algorithm, which worked as a graph-based ranking algorithm, automatically identified the most important sentences in the text (Mihalcea & Tarau, 2004). The adoption of TextRank is predicated on its inherent capabilities to efficiently extract pivotal sentences from voluminous textual datasets, making it particularly apt for the analysis of news commentary. The brief steps are as follows:

Document segmentation: The given document D is divided into n sentences S1, S2, …, Sn,  each sentence being a collection of words. Subsequently, a network graph G=V,E is constructed, where V is the set of sentences in the document; E is the set of edges between nodes, in the form Vi,Vj, representing an edge between nodes Vi and Vj. The similarity between sentences is calculated using the following formula as the weight of the edges: (4) Wij=similaritySi,Sj=wk,wk∈Si,wk∈Sj logSi+ logSj

where Wij represents the weight, or similarity, between sentences Si and Sj.

Weighted graph formation: With the inclusion of weights W, a weighted undirected graph G′=V,E,W is formed. The cumulative weight of each node is iteratively calculated as follows: (5) WSVi=1−d+d∗∑Vi∈InViWji ∑Vk∈OutVjWjkWSVj.

Wij denotes the weight between two nodes Vi and Vj, correlating to the similarity between sentences Si and Sj; The notation InVi denotes the collection of edges directed towards node Vi, while Out(Vj) denotes the collection of edges emanating from node Vj, with Vi and Vj representing the node under calculation and the node to be shared, respectively. Consequently, WS(Vi) refers to the weight of the node under calculation, whereas WS(Vj) denotes the weight of the node to be shared. The term d serves as the damping factor, signifying the probability of transitioning from a given node to any other node within the network.

Convergence and summary selection: The iterative process continues until the scores stabilize (convergence), after which sentences are ranked based on their final scores. The top k sentences, where is a predefined number, are selected to compose the summary, effectively capturing the document’s key points.

As a result, this feature automatically extracted crucial, representative information from complex international news commentary to form the summary of the article. This automated summarization method enabled users to quickly grasp the core information of the article without having to read the entire content, saving time and providing convenience. It was especially vital in the era of overwhelming information overload and improved the efficiency of information retrieval for users.

Implementation of text sentiment analysis

The system utilized NLTK’s sentiment analysis tool Valence Aware Dictionary for sEntiment Reasoning (VADER), to perform sentiment analysis on the preprocessed text (Fig. 6). VADER relied on a sentiment lexicon that contained a large collection of common English words, associated sentiment polarity, and intensity scores (Elbagir & Yang, 2019). It can recognize negation words and degree adverbs, taking into account the context of words in the text and their relationships to each other. VADER calculated a comprehensive sentiment score that categorized the sentiment as positive, neutral, or negative based on this score.

Figure 6 Sentiment analysis.

In this study, a standardized threshold for classifying sentences as positive, neutral, or negative was set as follows (Yin et al., 2022): Sfi=positive vscore>=0.05negative vscore<=−0.05neutral otherwise

Where vscore was the composite score of the i-th news, Sfi was the final polarity of the tweet. If the composite score was not less than 0.05, the sentence was considered positive. If the score was not greater than −0.05, its polarity was negative. Otherwise, the sentence polarity was neutral.

Implementation of keyword extraction

In Python, keyword extraction can be accomplished by using the Natural Language Toolkit (NLTK) (Loper & Bird, 2002) to calculate word frequencies and store the N most frequent words as keywords in a database (as shown in Fig. 7). The critical steps implemented for keyword extraction were as follows:

Figure 7 Keyword extraction.

The critical steps for keyword extraction were executed as follows: Firstly, the NLTK’s word_tokenize function was employed for tokenization, breaking down the text into individual words. Secondly, the FreqDist class from NLTK was used to calculate the frequency distribution of the words obtained post-tokenization, facilitating the handling of frequency distributions. Lastly, the most_common(N) function retrieved the top N words with the highest frequencies, returning them in descending order. These steps, when combined, enabled the effective extraction of keywords from the text data. The extracted keywords and their frequencies were then ready to be stored in a database for subsequent analysis or reference purposes.

Implementation of similarity recommendation analysis

In this paper, the system implemented a similar article recommendation feature by analyzing the similarity between articles. The system utilized natural language processing techniques such as word segmentation, TF-IDF (Ramos, 2003), and text similarity calculation methods such as cosine similarity (Rahutomo, Kitasuka & Aritsugi, 2012). Cosine similarity quantifies the similarity between two vectors in an n-dimensional space by computing the cosine of the angle between them. This measure is obtained by dividing the dot product of the vectors by the product of their magnitudes. (6) SimA→,B→=A→⋅B→A→B→=∑i=1nAiBi∑i=1nAi2∑i=1nBi2.

The cosine similarity value ranges from −1 to 1, where 1 means A→ and B→ are in the same direction, 0 indicates orthogonality (no similarity), and -1 means they are in opposite directions. In addition to its primary applications, the algorithm has found utility in a variety of other domains. The Cosine Similarity-based Image Filtering (CosSIF) algorithm has significantly enhanced medical image analysis models by improving the diversity and quality of training data (Islam, Zunair & Mohammed, 2024). Moreover, the cosine similarity matrix-based video scene segmentation (CSMB-VSS) algorithm has capitalized on the relationships between video scenes and shot similarities to significantly optimize video segmentation (Chen et al., 2024). Furthermore, the integration of cosine similarity with the Alternating Least Squares (ALS) algorithm has substantially enhanced the accuracy and performance of recommendation systems (Hasan & Ferdous, 2024).

By using these techniques, the system efficiently computed the similarity between many articles, which enabled the recommendation of similar articles (as shown in Fig. 8).

Figure 8 Similarity recommendation analysis.

Implementation of topic analysis

This system utilized the gensim’s corpora module and LDA algorithm (Yu & Yang, 2001) for topic analysis (as shown in Fig. 9), to efficiently parse and categorize the thematic elements of text data, with the following key steps:

Figure 9 Topic analysis.

Initially, the text data underwent word segmentation, wherein individual words were identified using a segmentation technique. Subsequently, a corpus and a corresponding dictionary were developed from this segmented data. The corpus served as a bag-of-words representation of the text, while the dictionary mapped words to their unique integer identifiers. The topic model was then trained utilizing the Latent Dirichlet Allocation (LDA) algorithm, a probabilistic method for identifying latent topics in collections of text. The LDA model is characterized by a distinct hierarchical architecture, which comprises three sequential layers: the corpus level, the topic level, and the level of feature words. This architecture is delineated in Fig. 10. Each document is assumed to be generated by a mixture of topics. This mixture is represented by a distribution θd over topics for document d, which is drawn from a Dirichlet distribution with hyperparameter α: θd∼Dirα. Each topic is characterized by a distribution over words. This distribution βk for topic k isalso drawn from a Dirichlet distribution with hyperparameter β: βk∼Dirβ. For each word in a document, a topic zd,n is chosen from a Multinomial distribution parameterized by θd, and then a word wd,n is chosen from a Multinomial distribution parameterized by βzd,n: Choose topic zd,n∼Multinomialθd, Choose word wd,n∼Multinomialβzd,n. The joint probability of the model’s variables can be expressed as: (7) pθ,z,w|α,β=pθ|α∏n=1Npzn|θpwn|zn,β.

Figure 10 The topological structure diagram of LDA model.

Furthermore, the ultimate goal of applying LDA is to infer the posterior distribution: (8) pz,θ|w,α,β.

However, due to its computational complexity, direct computation of this posterior is infeasible for non-trivial corpora. Thus, approximation techniques such as Variational Inference or Gibbs Sampling are employed to estimate the distribution of topics across documents and the distribution of words across topics. Finally, the identified topic keywords were stored in a database, facilitating the generation and output of automatic article summaries. This application of the LDA model directly facilitated the extraction of topic keywords, thereby improving the accessibility and relevance of the content analyzed.

Implementation of data visualization function

This article has successfully implemented data visualization analysis using JavaScript as the basis and integrating the Echarts plugin. To dynamically retrieve data from the database and display it on the web page, the system adopted the Flask backend framework and utilized Ajax asynchronous calls (Shi, Fan & Li, 2009). This technology enabled a more vivid and intuitive presentation of data and improved user interaction, which increased user engagement and satisfaction.

Implementation of comprehensive data analysis and visualization

The comprehensive analysis of international news commentary data (as shown in Fig. 11) mainly included the following aspects:

Figure 11 Comprehensive data analysis and visualization.

Proportional distribution of categories: This analysis showed the distribution of different categories in the international news commentary data, indicating the proportions of different themes or topics in the dataset.

International news commentary publication time distribution: This analysis illustrated the distribution of publication times for international news commentary, providing insights into the temporal patterns of content generation.

International news commentary publishing source distribution: This analysis illustrated the distribution of different publishing sources for international news commentary, indicating which platforms or media outlets were prevalent in the dataset.

Distribution of the number of international news commentary: This analysis examined the distribution of international news commentary written by individual critics, providing an overview of their writing frequency and contribution.

Implementation of visualized analysis for article details

The data visualization of the international news commentary content details mainly included the display of the sentiment trend in articles, the display of the abstract, the visualization of the extracted keywords, the display of the topic analysis, and the display of similar article recommendations. This part involved retrieving the article’s unique identifier (ID), which corresponded to the detail page when a user clicked on it. Based on this ID, the system visualized the content of the specific article (as shown in Fig. 12).

Figure 12 Visualized analysis for article details.

Implementation of data retrieval function

The implementation of this module was based on the international news commentary database of this system, and the specific workflow was as follows:

Upon activation of the search function by the user, the entered keyword was retrieved. This action triggered an Ajax request using jQuery, which was selected for its efficiency in handling asynchronous requests without needing to reload the web page, thus enhancing user experience. The request was then sent to the backend Flask application via the ’/api/search’ route. The Flask framework processed this request by extracting the keyword. Subsequently, the backend performed a multi-condition fuzzy query on the database using SQLAlchemy ORM, chosen for its robustness in handling complex queries and its compatibility with multiple database systems. This ORM framework conducted the search across multiple fields in the database, allowing for a broad and flexible matching process that was essential for effective search functionality. The query results were formatted into a list of dictionaries and sent back to the frontend in JSON format, completing the search and retrieval process.

The frontend rendered the search results based on the data returned by the backend. The title, abstract, author, source, and publication time of each comment were concatenated into an HTML string. This step was crucial as it transforms raw data into a user-friendly format. Finally, the rendered HTML string was inserted into the page, presenting the search results as a list (as shown in Fig. 13).

Figure 13 Data retrieval function.

Discussion and Conclusions

This article proposed an intelligent processing system for international news commentary data that improved the depth of analysis for complex commentary texts. This innovative system stood out from traditional platforms due to several key advantages:

•Enhanced depth of analysis. The system enabled in-depth analysis of complex international news commentary data. Its intelligent algorithms enabled a nuanced understanding of textual content, providing valuable insights into the attitudes and emotions of news commentary.

•Overcoming challenges. The system effectively addressed tackled issues, such as information overload, noise, and cultural barriers by providing user solutions.

•Intelligent processing. The proposed system optimized data processing efficiency, unlike traditional platforms. Using advanced algorithms, the system automated analysis, recommendation, and summary generation for international news commentary, enabling real-time analysis and decision-making.

While intelligent processing systems have significantly advanced the analysis of international news commentary, further enhancements in speed and efficiency are still essential. The potential integration of large generative models, such as GPT-4, could transform its capabilities, enabling the generation of more accurate and nuanced summaries and recommendations in real-time (Achiam et al., 2023). These models are particularly adept at handling complex contexts due to their extensive training on diverse datasets. Moreover, advancements in AI technologies can enhance real-time processing and adaptability, allowing for continuous updates and relevance as new data emerges. Additionally, incorporating an AI assistant feature could take the system beyond static analysis, enabling users to engage in direct dialogue, ask questions, and receive comprehensive, context-aware interpretations of international news commentary. Future enhancements should focus on leveraging these AI capabilities to improve operational efficiency and extend the system’s analytical depth (Casheekar et al., 2024). Consequently, continuous improvement will be essential for the system in the future, allowing it to address current and future challenges better and provide more accurate, in-depth, and comprehensive international news commentary analysis services.

The authors sincerely appreciate the help from the editor and the reviewers for their suggestions and comments.

Additional Information and Declarations

Competing Interests

Author Contributions

Data Availability

The authors declare there are no competing interests.

Shiting Liao conceived and designed the experiments, performed the experiments, analyzed the data, performed the computation work, prepared figures and/or tables, authored or reviewed drafts of the article, and approved the final draft.

Yunpei Wang conceived and designed the experiments, analyzed the data, prepared figures and/or tables, authored or reviewed drafts of the article, and approved the final draft.

Qingnian Wang conceived and designed the experiments, performed the experiments, analyzed the data, prepared figures and/or tables, authored or reviewed drafts of the article, and approved the final draft.

The following information was supplied regarding data availability:

The data and code is available at Zenodo: Shiting. (2023). International News Commentary Data Intelligent Processing System. Zenodo. https://doi.org/10.5281/zenodo.10212285.

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
