# Peer review of "Design and implementation of the international news commentary Data Intelligent Processing System"

_PeerJ Computer Science, doi:10.7717/peerj-cs.2376_

## Round 0.1 · original submission · Major Revisions

Based on the reviewers comments, you may have the opportunity to address the different issues raised. Please try to focus it on computer science field.

**Language Note:** The review process has identified that the English language must be improved. PeerJ can provide language editing services - please contact us at copyediting@peerj.com for pricing (be sure to provide your manuscript number and title). Alternatively, you should make your own arrangements to improve the language quality and provide details in your response letter. – PeerJ Staff

Reviewer 1 ·

Basic reporting

This paper has the following major concerns:

1) The author states that "international news commentary has changed." What does this mean, and how has it changed?

2) The author mentions that it causes noise. What is the meaning of noise in this context? Is it referring to Signal-to-Noise Ratio (SNR) or something else? Overall, the abstract is poorly written.

3) The author does not provide any details about their proposed work, methodology, model, etc., in the abstract.

4) Why is the related work divided into subsections? What is the criterion for this division?

5) The methodology does not provide insights into the proposed work, methods, framework, etc. Additionally, the figures are poorly drawn.

6) This paper does not provide insights into how the proposed work is efficient in its implications.

7) The writing needs a lot of improvements as well.

Experimental design

Included in Basic Reporting

Validity of the findings

Included in Basic Reporting

Additional comments

Nil

·

Basic reporting

### English language

English language is clear an professional.

### Intro, background and references

Abstract and introduction are fully focused on the application of the described method. They do not discus the natural language processing method that should be in the focus of the text, if it has to be published in a computer science journal.

Keywords that are selected by authors either are not appropriate, or they show that this text must be addressed to a journal that is not focused on computer science. An exception is the keyword *Python* which is not appropriate for this text, because the text does not suggest anything that is related to the programming language Python as an object of research.

### Text structure

Needs a general restructure. It contains too much bullets, some of them not in a standard form.

### Figures and tables

The proposition contains quite few figures that describe the proposed method and too much screenshots. Some of the figures contain inscriptions with small font that are hard to read.

### Raw data

The authors submitted implementation written in Python. I would like to point out that this does not mean that *Python* must be included as a keyword, since the text does not discuss anything that is related to a research in the Python language.

Experimental design

### Originality of the research and scope of the journal

The text is heavily focused on the application itself. Like this, it is more appropriate as a proposition that is outside computer science scope.

### Research questions definition

Research questions are not well defined. There are claims which have no proof like even in the abstract:

> Traditional platforms often lack data-driven analysis capabilities.

There must be a real analysis what are the gaps in the online systems and how an NLP system can fill them.

### Technical and ethical standards of the research

From a technical point of view, the text does not contain a description of a single method, model or algorithm. Technically, this cannot be a computer science article.

### Description detail sufficiency to replicate

I don't see an algorithm or method described in this text.

Validity of the findings

The section **Experiments and Results** actually gives an implementation summary that discusses different technologies used by authors. I would like to stress again that list of technologies or programming language selection does not turn a research into computer science article proposition.

### Impact and novelty

There is no comparison with existing similar systems. No impact nor novelty can be concluded from this proposition.

Additional comments

The proposition in computer science journal must put in the center of the description the methods and algorithms developed. It must contain the mathematical description of the NLP methods that are used. It must describe in detail developed methods (using mathematics and pseudocode, etc.). It must contain analysis for the effectiveness of the proposed approach based on deep mathematical analysis of the proposed algorithms, or based on experimental comparison with existing methods, or both. In this shape, this text is not in the scope of computer science.

---

## Round 0.2 · Major Revisions

Based on the Reviewer 2, the authors must detail the method, and clarify the scope of the manuscript in the computer science field. Also, the language must be proofread.

**Language Note:** The Academic Editor has identified that the English language must be improved. PeerJ can provide language editing services - please contact us at copyediting@peerj.com for pricing (be sure to provide your manuscript number and title). Alternatively, you should make your own arrangements to improve the language quality and provide details in your response letter. – PeerJ Staff

Reviewer 1 ·

Basic reporting

The authors have revised the paper based on previous listed concerns, therefore, i am happy to suggest the acceptance of the article.

Experimental design

No further concerns

Validity of the findings

No further concerns

·

Basic reporting

### English language

English language is clear, but the style does not answer academic standards. There are expressions like:
> When a user clicked the search button...

### Intro, background and references

Abstract and introduction focus the submitted text in the scope of the application of the topic, not in the scope of the development of the method.

### Text structure

Needs a general restructure. It contains too much bullets, some of them not in a standard form.

### Figures and tables

As I mentioned before, most of the figures are screenshots that bring no information about any method in the scope of the computer science. An exception is Figure 1 that shows the workflow of a web-page. The figure is barely explained in the text.

### Raw data

The authors submitted implementation written in Python.

Experimental design

### Originality of the research and scope of the journal

The text is heavily focused on the application itself. Like this, it is more appropriate as a proposition that is outside computer science scope.

### Research questions definition

The text does not contain a clear definition of a problem in the scope of informatics or computer science that is analyzed and solved in the proposition.

### Technical and ethical standards of the research

From a technical point of view, the text does not contain a description of a single method, model or algorithm. Technically, this cannot be a computer science article.

### Description detail sufficiency to replicate

I don't see an algorithm or method described in this text.

Validity of the findings

The section **Experiments and Results** contains description of a series of actions taken by the system. The text does not clearly state an algorithm, but a sequence of technical actions.

### Impact and novelty

No impact nor novelty can be concluded from this proposition.

Additional comments

The proposition in computer science journal must put in the center of the description the **methods and algorithms** developed.

---

## Round 0.3 · Major Revisions

Based on the reviewer comments, the manuscript must be revised before acceptance.

Reviewer 1 ·

Basic reporting

The authors addressed previous comments, hence this paper is recommended for publication.

Experimental design

N/A

Validity of the findings

N/A

Additional comments

N/A

Reviewer 3 ·

Basic reporting

The article "Design and Implementation of the International news
commentary Data Intelligent Processing System" has a good theme and can contribute to the magazine and readers.

Some suggestions for improvement

In the introduction, include new features that this work presents. I suggest a 2024 citation.

In related work
It is imperative to include current references and literature (2022, 2023, and 2024), it is not clear what criteria and comparison parameters were used between the literature researched about this work.
I strongly suggest including a comparison table.

Experimental design

It is not clear in the methods which variables were used and their respective justifications. However, there is a link to the data set provided by the authors at https://zenodo.org/records/10212285
It is unclear to the reader which and how the algorithms were defined, especially how they can be used in other scenarios and applications.

The mathematical equations are not clear about their use and adaptation to the proposed solution, as it stands it seems more like they were placed to provide focus and mathematical credibility but without justification and concrete evidence.

Validity of the findings

In the conclusions, the problems and results obtained, news, and especially the scientific contributions are not evident. Nor is it clear what future work and the exact points that future researchers will be able to continue from this work are.

Additional comments

References
Review, some are outdated or contain insufficient information, including DOI or ISSN.

The figures included in the annexes leave the work poor, the quality and way in which they are presented are very poor.

I believe it is a work in progress and therefore, I suggest that the authors improve it considerably and resubmit it.

I hope to have contributed to improving the quality of the work developed.

---

## Round 0.4 · Major Revisions

Dear authors, please accept my apologies, please address the comments of the reviewer

Reviewer 3 ·

Basic reporting

The article "Design and implementation of the international news
commentary Data Intelligent Processing System" does not describe or show how the implementations were carried out or the variables used. Tests and results are based on a few figures with few details.

Experimental design

I did not identify concrete evidence of criteria and parameters used, algorithm developed, or justifications for implemented code.
I did not understand which problems were solved.

Validity of the findings

What is new about the work?
What characteristics justify publishing the results presented? What results?

If we forget the technical part, there is also no comparison with the literature, robust related works, or presentation of new features of the work in relation to the current bibliographic state of the art.

Additional comments

I believe that authors should reflect on what concrete implementations their work has? What technical tests and comparisons?

What are the new developments?

What scientific problems does it propose to solve and have it actually managed to solve?

What scientific contributions does the work present?

---

## Round 0.5 · accepted · Accept

Dear authors,

Despite the comments of Reviewer 3, the overall feedback and revisions to your paper have been discussed with the Section Editors and we believe the manuscript was carefully revised and so it can be accepted as it is.

Best regards

Reviewer 3 ·

Basic reporting

The authors made modifications and changes to the work.

Experimental design

This topic does not yet have consistent contributions and concrete evidence, it needs to be detailed in depth with specifications and better details.

Validity of the findings

Contribution and novelty presentation are weak, and few research problems are solved concretely.

Additional comments

Technically, the work is weak; the authors could explore this aspect better. Theoretically, the work must be expanded, compared, and present concrete evidence.